# The Anticancer Potential of Kaempferol: A Systematic Review Based on In Vitro Studies

**DOI:** 10.3390/cancers16030585

**Published:** 2024-01-30

**Authors:** Everton Freitas de Morais, Lilianny Querino Rocha de Oliveira, Hannah Gil de Farias Morais, Maurília Raquel de Souto Medeiros, Roseana de Almeida Freitas, Camila Oliveira Rodini, Ricardo D. Coletta

**Affiliations:** 1Graduate Program in Oral Biology, Department of Oral Diagnosis, School of Dentistry, University of Campinas, Piracicaba 13414-018, SP, Brazil; evertonf@unicamp.br (E.F.d.M.); l265902@dac.unicamp.br (L.Q.R.d.O.); 2Postgraduate Program in Oral Science, Federal University of Rio Grande do Norte, Natal 59000-000, RN, Brazil; hannah.gil.102@ufrn.edu.br (H.G.d.F.M.); maurilia.souto.088@ufrn.edu.br (M.R.d.S.M.); roseana.freitas@ufrn.br (R.d.A.F.); 3Department of Biological Sciences, Bauru School of Dentistry, University of São Paulo, Bauru 17012-901, SP, Brazil; carodini@usp.br

**Keywords:** natural products, flavonoids, kaempferol, therapeutic profile, chemoagent, anticancer

## Abstract

**Simple Summary:**

Kaempferol, a natural compound commonly found in fruits, vegetables, and plants, has gained interest within the scientific community because of its anticancer properties against different types of tumors. The results of this review reveal that kaempferol exerts anticancer effects on many types of tumor cells by different mechanisms, providing evidence of its potential as a cancer drug.

**Abstract:**

Given the heterogeneity of different malignant processes, planning cancer treatment is challenging. According to recent studies, natural products are likely to be effective in cancer prevention and treatment. Among bioactive flavonoids found in fruits and vegetables, kaempferol (KMP) is known for its anti-inflammatory, antioxidant, and anticancer properties. This systematic review aims to highlight the potential therapeutic effects of KMP on different types of solid malignant tumors. This review was conducted following the Preferred Reporting Items for Systematic Review and Meta-Analyses (PRISMA) guidelines. Searches were performed in EMBASE, Medline/PubMed, Cochrane Collaboration Library, Science Direct, Scopus, and Google Scholar. After the application of study criteria, 64 studies were included. In vitro experiments demonstrated that KMP exerts antitumor effects by controlling tumor cell cycle progression, proliferation, apoptosis, migration, and invasion, as well as by inhibiting angiogenesis. KMP was also able to inhibit important markers that regulate epithelial–mesenchymal transition and enhanced the sensitivity of cancer cells to traditional drugs used in chemotherapy, including cisplatin and 5-fluorouracil. This flavonoid is a promising therapeutic compound and its combination with current anticancer agents, including targeted drugs, may potentially produce more effective and predictable results.

## 1. Introduction

Cancer comprises a group of malignant disorders that develop in different tissues and organs and are characterized by uncontrolled growth of altered cells and spread to distant sites. Cancer is a leading cause of death worldwide. According to the 2020 GLOBOCAN online database report, the annual number of cases is estimated to increase from 19.3 million in 2020 to 28.4 million in 2025 (an increase of 47% compared to 2020) [1]. The main cancer treatments are surgery, radiotherapy, and chemotherapy; however, additional strategies are applied to some tumors, such as hormonal therapy or photodynamic therapy [2]. Although biological treatments, including targeted therapy and immunotherapy, have changed the face of cancer therapy, they are constantly being improved and evaluated [2,3]. So far, all therapeutic options are associated with important side effects, in addition to the development of resistance to antineoplastic drugs [4]. In contrast, natural compounds are increasingly being used for the prevention and treatment of cancer, with a reduced risk of adverse reactions [5].

Natural compounds derived from plants (phytochemical compounds) are particularly interesting and have been widely used in the treatment of different pathological processes such as cancer because of their increased bioavailability and better tolerability compared to synthetic drugs [6]. Furthermore, recent studies indicate that a variety of phytochemicals can sensitize tumor cells to antitumor drugs, reversing tumor resistance and decreasing toxic effects in different malignant neoplastic processes [7,8]. Flavonoids are polyphenolic compounds synthesized as secondary bioactive metabolites that are responsible for the color, flavor, and pharmacological activities of plants [9]. Kaempferol (KMP) (3,5,7-trihydroxy-2-(4-hydroxyphenyl)-4H-1-benzopyran-4-one) (Figure 1), a flavonoid widely distributed in a variety of vegetables, fruits, and medicinal plants, has shown antioxidant and anti-inflammatory activity [10]. KMP has been suggested to reduce the risk of cardiovascular and neuroinflammatory diseases, and recent studies indicate it as a promising antineoplastic agent [11,12,13]. Within this context, KMP has been shown to regulate different mechanisms involved in oncogenesis and to sensitize neoplastic cells to chemotherapeutic agents [14,15,16]. Furthermore, KMP can induce mechanisms that disrupt the survival of tumor cells [14,17].

The literature exploring KMP potential in cancer treatment is extensive, but no systematic synthesis of in vitro studies elucidating the molecular mechanisms and pathways targeted by KMP is available. In this systematic review, we sought to summarize the available literature on the in vitro anticancer effects of KMP, discussing the pathways and molecular mechanisms regulated by KMP and highlighting its potential as a natural anticancer drug. We also critically analyzed the gaps in knowledge that limit the use of KMP in cancer treatment.

## 2. Materials and Methods

This systematic review followed the Preferred Reporting Items for Systematic Reviews and Meta-Analyses (PRISMA) guidelines (Appendix A Appendix A) [18]. The PICO criteria were used to develop the research questions: population—cell lines derived from malignant solid tumors; intervention—treatment with KMP; comparison—control group not treated with KMP; outcomes—analysis of the behavior of the neoplastic cell after treatment with KMP using functional assays (migration, invasion, viability, proliferation, sensitization to chemotherapy).

The research questions were: “Is KMP a potential anticancer agent?”, “What are the molecular mechanisms regulated by KMP in malignant neoplastic cell lines?” and “Does KMP exert a modulatory effect and act as a potential antineoplastic agent?”. The study protocol was registered in the Open Science Framework: https://osf.io/z5nsg/ (accessed on 27 January 2024).

### 2.1. Search Strategy

To identify primary research articles that evaluated the in vitro effects of KMP on cancer cells, we searched the EMBASE, Medline/PubMed, Cochrane Collaboration Library, Science Direct, Scopus, and Google Scholar databases. In addition, the references of pre-selected articles were hand-searched. Searches were performed in December 2022. However, a second literature search was performed at the beginning of September 2023, retrieving articles published between January and August 2023. The search strategy was based on combinations of the following keywords: (“cancer cell” [MeSH] AND “in vitro studies” [MeSH] AND “kaempferol” [MeSH] AND “biological behavior” [MeSH] AND (“humans” [MeSH]). The complete search strategy is given in Appendix B.

### 2.2. Study Selection and Selection Criteria

Initial screening based on titles and abstracts was performed by two independent reviewers who classified the studies as “yes”, “no”, or “maybe” based on inclusion criteria. Studies classified as “yes” or “maybe” were selected for full-text reading. The eligibility criteria were applied in both steps.

Articles that evaluated the effects of KMP treatment on the biological behavior of malignant solid tumor cells were selected for the present systematic review. The search was performed without time or language restrictions. The following studies were excluded from this systematic review: (i) studies that used only non-cancer cells; (ii) in vivo studies only; (iii) studies that did not evaluate KMP or did not evaluate it alone; (iv) studies that used only in silico analysis or bioinformatics. Two reviewers independently selected the articles and any disagreement was resolved by consensus.

### 2.3. Data Extraction, Analysis, and Risk of Bias Assessment

The authors independently extracted the following data from the included studies using a pre-established form: authors, year of publication, country, tissue origin of cell lines, origin/extraction method of KMP, dosage of KMP, treatment time, effects on biological behavior, effects on expression of biomarkers, effects on cell sensitization to chemotherapy, and main findings of study. The results of the individual studies were then summarized, categorized, pooled, and analyzed. The quality of evidence was assessed using the Grading of Recommendations Assessment, Development, and Evaluation (GRADE) criteria [19,20], which were adapted for in vitro studies as described by Pavan et al. [21]. The quality of evidence of the articles was also classified as high, moderate, low, or very low.

## 3. Results

### 3.1. Study Characteristics

A total of 1238 articles were retrieved; after the removal of duplicates, 231 articles remained for the first screening based on titles and abstracts. One hundred and twenty-six articles were selected for the next phase. After full-text reading, 64 articles were included in the qualitative synthesis [10,12,14,17,22,23,24,25,26,27,28,29,30,31,32,33,34,35,36,37,38,39,40,41,42,43,44,45,46,47,48,49,50,51,52,53,54,55,56,57,58,59,60,61,62,63,64,65,66,67,68,69,70,71,72,73,74,75,76,77,78,79,80,81]. The list of excluded studies and reasons for exclusion are shown in Appendix C. Figure 2 depicts the flowchart of the study screening and selection process.

The selected studies were published between 2003 and 2022 and were all written in English. The studies were conducted in 14 different countries, 25 (38.5%) of them by Chinese groups. The KMP dosage, the duration of treatment, and the cell lines used in the selected studies varied. The main features and findings of the studies are presented in Appendix D.

### 3.2. KMP Inhibits Cell Proliferation, Invasion, and Migration and Promotes Cell Death

KMP exhibits notable properties in combating cancer. Our systematic review indicated that this compound exerts inhibitory effects on cell proliferation, disrupting the cell cycle at key stages and halting cell division. Additionally, KMP was associated with increased cytotoxicity in neoplastic cells, reinforcing its potential as a therapeutic agent against cancer development and progression. KMP significantly affected the migration and invasion of neoplastic cells, directly interfering with cancer dissemination. These properties highlight the potential of KMP as a promising option in the pursuit of effective therapeutic strategies to treat cancer (Figure 3). Table 1 shows the main findings according to type of tumor.

Most of the studies included in the present review explored the effects of KMP on tumor cell death [10,12,22,27,28,29,30,31,32,35,36,40,46,51,52,53,55,56,57,62,63,66,70,74,75,76,77,80,81]. These studies reported the apoptosis-inducing properties of KMP, which can be partially attributed to its impacts on the MAPK pathway. In human lung cancer cells (A549 cell line), activation of the mitogen-activated protein kinase (MAPK) pathway is a key factor in KMP-induced apoptosis [22]. Furthermore, KMP was able to reduce B-cell lymphoma 2 (BCL2) levels and increase the expression of tumor suppressor proteins such as p53 and BCL2-associated agonist of cell death (BAD) [40]. Protein p53 plays a key role in the repair of damaged DNA. According to Luo et al. [40], KMP prevents the phosphorylation of protein kinase B (PKB, also known as AKT) and upregulates p53 expression, inducing apoptosis of ovarian cancer cells. In addition to apoptotic effects, KMP also has antioxidant activity. Studies have shown that KMP reduces the production of free radicals and other products such as reactive oxygen species (ROS) in different neoplastic cell lines, increasing the expression of manganese superoxide dismutase, a mitochondrial antioxidant enzyme [10].

Twelve of the studies included in this systematic review explored the role of cellular toxicity of KMP in lung cancer cells using A549 [22,44,47,48,49,53,70,73,80] or H460 cells [27,60,61]. The findings were consistent, indicating a strong cytotoxicity of KMP. Another focus of studies on KMP was ovarian cancer [34,37,40,41,57,60,79]. The results demonstrated positive effects, with KMP reducing cell viability and angiogenesis and increasing apoptosis, with cell cycle arrest at G2/M. Luo et al. [34] suggested that KMP inhibits tumor angiogenesis in human ovarian cancer by suppressing vascular endothelial growth factor (VEGF) expression through a hypoxia-inducible factor (HIF)-dependent (AKT/HIF) pathway in ovarian cancer cells. In cervical cancer, KMP has been shown to reduce cell viability and increase apoptosis [55]. Two studies included in this systematic review evaluated the effects of KMP in bladder cancer, with both reporting consistent findings of increased apoptosis, increased cytotoxicity, induction of S-phase cell cycle arrest, and reduced cell proliferation, motility, and invasion [46,62].

Several studies have shown that KMP is strongly involved in the control of cell cycle arrest [29,43,45,49,51,56,57,62,63,65,77,81]. Gao et al. [57], investigating the effects of KMP on the cell cycle and extrinsic apoptosis of ovarian cancer cells, demonstrated that KMP induced G2/M cell cycle arrest via checkpoint kinase 2 (CHK2)/cell division cycle 25C (CDC25C)/cyclin-dependent kinase 2 (CDC2) and CHK2/p21/CDC2 in A2780/CP70 human ovarian cancer cells. The authors further showed that KMP stimulates apoptosis triggered by Fas-associated death domain protein (FADD) and caspase-8, suggesting that CHK2 and death receptors play important roles in the anticancer activity of KMP in A2780/CP70 cells [57]. Corroborating these findings, KMP treatment increased p53 levels in MDA-MB-453 breast cancer cells, which led to G2 cell cycle arrest [29]. Luo et al. [37] reported that the increased c-Myc levels after KMP treatment antagonized cyclin-dependent kinase inhibitor 1A (CDKN1A) mRNA expression, interrupting cell cycle progression by inhibiting the activity of cyclin-dependent kinases. KMP also yielded positive results in colon cancer, inhibiting cyclin D-dependent kinase 2 (CDK2) and cyclin D-dependent kinase 4 (CDK4) activities and inducing cell cycle arrest both at G1 (after 6 h of treatment) and G2/M (after 12 h) [43], as well as reducing glucose consumption [76], cell viability [24] (Liu et al., 2019), proliferation [14,45,70], and invasion and migration [59]. Zhang et al. [81], analyzing three different prostate cancer cell lines (22Rv1, PC-3, and DU145), showed that KMP exerts similar effects in terms of reducing cell proliferation but at different points depending on the cell line, arresting the cell cycle at G1 in 22Rv1 and at S/G2 in PC-3. Similar results were reported by Campbell et al. [25] and Yoshida et al. [32]. In the G1 phase, there is a crucial checkpoint where the cell assesses its environment and internal conditions before committing to DNA synthesis and cell division. By arresting the cell cycle at G1, KMP may be preventing the progression of these cells into the synthesis (S) phase, thereby inhibiting DNA replication and subsequent cell division. By affecting the cell cycle at the S/G2 transition, KMP may be interfering with DNA replication and the preparation for cell division. Understanding how a substance like KMP affects the cell cycle at different points in cancer cell lines is crucial for developing targeted and effective therapies for cancer. This tailored approach could potentially enhance therapeutic outcomes by exploiting the unique vulnerabilities of specific cancer cell types.

Zhu et al. [65] demonstrated that the inhibitory effects of KMP on proliferation are greater in estrogen receptor-positive breast cancer cells (BT474 cells) compared to estrogen receptor-negative cells (MDA-MB-231 cells). Furthermore, the authors observed that KMP contributed to the induction of G2/M arrest, apoptosis, and DNA damage in MDA-MB-231 cells. Oh et al. [26] showed that KMP blocks estradiol-induced tumor formation, confirming that this flavonoid can inhibit the malignant transformation caused by estrogens. The authors suggested that KMP may ensure an adequate level of estrogenic activity in the body, potentially exerting beneficial effects on diseases caused by estrogen imbalance such as breast cancer. The study by Abdullah et al. [64] evaluated the effects of KMP on neuroblastoma and reported an increase in neuroblastic differentiation and apoptosis and a reduction in cell viability and proliferation. Inden et al. [71] found that KMP provided significant protection against α-Syn-related neurotoxicity and induced autophagy through an increase in lysosome biogenesis by inducing transcription factor EB (TFEB) expression and reducing the formation and accumulation of α-Syn amyloid fibrils, thus preventing the death of neuronal cells. Complementary results were obtained by Siegelin et al. [31] and Chen et al. [66] who described higher rates of autophagy and apoptosis and lower cell proliferation in glioblastomas. Gastrointestinal cancers have also been studied and the results support and encourage the use of KMP to reduce cell proliferation [23,53] and colony formation [28] and to increase cell cycle arrest at G2/M, autophagy, and cell death [51,58]. Only the study conducted by Liu et al. [74] reported the effects of KMP on gallbladder cancer, with favorable results that included increases in apoptosis and DNA damage and a reduction in cell viability, invasion, and migration. For pancreatic cancer, five studies investigated the effects of KMP using in vitro functional assays. The results revealed reduced viability, proliferation, and migration of both MIA PaCa-2 and Panc-1 cells, as well as increased apoptosis [33,52,60,78].

KMP showed inhibitory effects against aryl hydrocarbon receptor (AHR)- and nuclear factor erythroid 2-related factor 2 (Nrf2)-induced expression of the drug-metabolizing enzyme in hepatocellular carcinoma [68]. KMP’s inhibitory effects on AHR- and Nrf2-induced expression of drug-metabolizing enzymes suggest that KMP interferes with the activation of these pathways, which is relevant in the sense that the dysregulation of AHR and Nrf2 pathways has been associated with the promotion of tumor growth and resistance to chemotherapy in certain cancers, including hepatocellular carcinoma. In addition, the compound increased autophagy and reduced cell viability [39], proliferation [54], and migration and invasion [72]. Li et al. [28], Nair et al. [69], Wang et al. [44], and Yang et al. [77] investigated the effects of KMP in liver cancer. The authors reported positive results, including reduced colony formation, cell proliferation, migration, and invasion and increased cell viability, apoptosis, and cell cycle arrest in the G1 phase. Huang et al. [35] studied the anticancer effects and molecular mechanisms of KMP in human osteosarcoma cells and demonstrated that KMP significantly reduced the viability of U-2 OS, HOB, and 143B cells in a dose-dependent manner, with low cytotoxicity on hFOB cells, a human fetal osteoblast progenitor cell line. In vitro assays confirmed the effects of DNA damage, apoptosis in U-2 OS cells, increased cytoplasmic levels of calcium ions, and decreased mitochondrial membrane potential [35]. In additional experiments, Chen et al. [42] later showed that KMP decreased the DNA-binding activity of activator protein-1 transcription factor (AP-1), suggesting a potential role of the compound in the treatment of osteosarcoma metastasis. Dysregulated AP-1 activity is related with the development and progression of several cancers, and a reduction in the AP-1 DNA-binding activity promoted by KMP implies an interference in the events controlled by AP-1. Skin and oral cavity squamous cell carcinoma cell lines were also studied in the articles included in this systematic review, with KMP inducing cell cytotoxicity [60] and apoptosis and reducing proliferation [36] and colony formation [28]. Similar results were found for the treatment of melanoma with KMP, characterized by a reduction in colony formation, viability, and migration and an increase in apoptosis and cell cycle arrest at G2/M [28,63]. Zheng et al. [17] also highlighted the reduction in aerobic glycolysis in melanoma cells after treatment with KMP.

Wang et al. [75] demonstrated that KMP induces ROS-dependent apoptosis in pancreatic cancer cells through transglutaminase 2-mediated AKT/mammalian target of rapamycin (mTOR) signaling. The study by Chen et al. [66] points out that KMP increases ROS and decreases mitochondrial membrane potential in glioma cells. High levels of ROS induce autophagy and ultimately trigger glioma cell pyroptosis. The paradoxical behavior of cancer cells is intriguing. These cells redirect metabolic processes towards utilizing glutamine in the tricarboxylic acid cycle to mitigate cell damage induced by ROS and to maintain a stable cellular redox balance. ROS exert a dual role, acting either as harmful agents or as beneficial molecules that regulate cellular functions or induce cytotoxic effects depending on their concentration and duration. It is, therefore, crucial to devise strategies that harness cellular redox signaling in cancer, aiming to develop effective anticancer therapies or to target regulators of ROS for improved cancer treatments.

### 3.3. KMP Modulates the Expression of Cancer Biomarkers

Although KMP has been extensively studied as a natural flavonoid with cytotoxic effects against tumor cells, its effects on other carcinogenic processes have also been demonstrated. One step necessary for tumor cell migration and invasion is the acquisition of cell plasticity promoted by epithelial–mesenchymal transition (EMT). Within this context, many of the studies included in this review demonstrated the effects of KMP on EMT markers (Figure 4) [47,48,79]. KMP was able to reverse the EMT process in gastric, ovarian, and breast cancer cells by increasing E-cadherin expression and reducing the expression of mothers against decapentaplegic homolog 2 (Smad2), mothers against decapentaplegic homolog 4 (Smad4), transforming growth factor-β1 (TGF-β), N-cadherin, vimentin, and Snail [79]. EMT also depends on the expression of several proteases, including matrix metalloproteinases (MMPs). Ju et al. [72] evaluated the effect of KMP on hepatocellular carcinoma cells and the results showed that MMP-9 was dramatically decreased after KMP treatment. KMP also suppressed the phosphorylation of AKT, a key component of cell growth and survival, and has been associated with cell migration and adhesion through regulation of MMP-9.

KMP is considered to be a potent anti-inflammatory and antiangiogenic agent. The anti-inflammatory effects of KMP are primarily facilitated by the downregulation of numerous sequence-specific DNA binding factors, such as signal transducer and activator of transcription (STAT) [24], which induce the activation of inflammatory cytokines. Neoplastic cells also need nutrients and oxygen to survive. The main mediator of angiogenesis is VEGF. Evaluating the effect of KMP on ovarian cancer cell lines, Luo et al. [34] observed a significant reduction in VEGF and HIF-1, which are critical for adaptive vascular responses to ischemia/hypoxia. By targeting these key mediators, KMP may disrupt the angiogenic process, thereby limiting the blood supply to the tumor and impeding its growth.

### 3.4. KMP Sensitizes Cancer Cells to Chemotherapy

The studies included in this review also investigated the effects of the combination of KMP with antineoplastic drugs, including cisplatin [40], 5-fluorouracil (5-FU) [12,16,33], sorafenib [69], doxorubicin [77], and erlotinib [78].

Cisplatin is a commonly used alkylating chemotherapeutic agent that is known to induce cell death in different neoplastic processes. In the study conducted by Luo et al. [40], the combined administration of KMP and cisplatin reduced c-Myc mRNA concentration and increased CDKN1A mRNA levels in ovarian cancer cell lines, potentiating cell death. 5-FU is an antimetabolite chemotherapeutic agent widely used in the treatment of different types of cancers. Two studies included in this review evaluated its effect in combination with KMP [12,33]. In the study by Li et al. [12], the combination decreased the viability of colorectal cancer cells, with 5-FU inducing apoptosis potentiated by KMP. The combination of KMP and 5-FU was found to be more effective in promoting cell viability than either agent alone. The authors also observed the upregulation of proteins associated with apoptosis, such as BCL2-associated X (BAX), BCL2, and thymidylate synthase [12]. Combining low doses of KMP and 5-FU exerted an additive effect on the inhibition of MIA PaCa-2 pancreatic cancer cells [33]. In colorectal cancer cells, KMP overcame 5-FU resistance by regulating the microRNA-326–heterogeneous nuclear ribonucleoprotein A1/A2/polypyrimidine tract-binding protein 1–pyruvate kinase M2 (miR-326–hnRNPA1/A2/PTBP1–PKM2) axis [16].

Sorafenib, a multikinase inhibitor used in the treatment of hepatocellular carcinoma and other types of tumors, is associated with severe toxicity, which limits its utilization [69]. Nair et al. [69] combined sorafenib and KMP and found higher cytotoxicity at lower concentrations of sorafenib when combined with KMP than when used alone. KMP exhibited inhibitory effects on different liver cancer cell lines in a time- and dose-dependent manner and was not toxic to normal hepatocytes. Furthermore, KMP acted synergistically with the antitumor antibiotic doxorubicin, increasing its antineoplastic capacity [77]. Finally, Zhang et al. [78] observed that KMP potentiates the sensitivity of pancreatic cancer cells to the epidermal growth factor receptor (EGFR) tyrosine kinase activity inhibitor erlotinib, inhibiting cell proliferation and inducing apoptosis, and, thus, suggested KMP to be a valuable candidate for potentiating the effects of erlotinib.

### 3.5. Risk of Bias

Based on the GRADE assessment, 20 articles were classified as carrying a high overall quality of evidence, 43 articles were classified with moderate quality of evidence, and 1 article was classified as low quality (Appendix E). The main issues that drove to downgrading were related to indirectness, since 5 articles did not evaluate the effects of KMP in all cell lines of the study, and imprecision, as 41 articles did not calculate and adopt IC_50_ for treatments with KMP. IC_50_ is the most widely used and informative measure of a compound/drug’s efficacy, and its proper determination facilitates the development of useful methods for drug discovery and clinical tests [82]. Moreover, it allows a clear view of drug effects, and IC_50_ at a low concentration is prone to cause lower systemic toxicity when given to patients for treatment [83].

## 4. Discussion

Emerging data highlight the health-promoting properties of plant-derived flavonoids such as KMP. This tetrahydroxyflavonoid, in which four hydroxyl groups are located at positions 3, 5, 7, and 4′ [84], has received much attention over recent years as a major active flavonoid found in several medical plants, especially because of its anticarcinogenesis effects [85]. Our systematic review indicates that KMP exerts in vitro inhibitory effects on several carcinogenic pathways in different tumors, such as breast, prostate, liver, head, and neck cancers (Appendix D). KMP not only promotes the death of neoplastic cells but also inhibits proliferation, migration, invasion, angiogenesis, and EMT. Furthermore, KMP sensitizes tumor cells to chemotherapy with different drugs.

The preventive role of KMP in the development of cancer may be due, among other factors, to the elimination of ROS species. Elevated ROS production is generally considered a pro-tumoral factor, leading to DNA, protein, and lipid damage, improving adaptations to hypoxia, and promoting genetic instability [86,87]. However, in pancreatic cancers, ROS have been shown to act as a double-edged sword, either facilitating cancer progression or dramatically enhancing cell death [88,89]. ROS-mediated anticancer properties, including the inhibition of cell proliferation, cell cycle arrest, induction of apoptosis, and suppression of cell invasion and migration, have been reported for KMP in several types of cancer such as colorectal cancer, breast cancer, melanoma, and hepatocarcinoma [29,53,90]. For example, KMP triggers ROS-induced apoptosis in human breast cancer via the c-Jun N-terminal kinase (JNK) signaling pathway [91] and potentiates apoptosis in human hepatoma cells through the mitochondrial apoptotic pathway regulated by p53-induced gene 3 (PIG3) [90]. In vitro experiments using cells from squamous cell carcinomas of the oral tongue (SCC-25, SCC-4, and SCC-1483), esophagus (Eca-109), oral cavity (PCI-13), and hypopharynx (FaDu) demonstrated the antiproliferative effects of KMP [36,92,93]. The expression of MMP-2, Bcl-2, c-Jun (AP-1 transcription factor subunit), and hexokinase-2 was reduced by KMP, which contributed to cell cycle arrest in the G0/G1 phase. In a mouse xenograft model, the ability of KMP to successfully inhibit tumor growth through loss of hexokinase-2 expression and EGFR activity in cancerous tissues provided further evidence of the anticancer efficacy of the drug. KMP has also been shown to suppress the invasion and migration of SCC-4 cells, decreasing enzyme activity and MMP-2 gene expression. Furthermore, KMP inhibits MAPK1/2 phosphorylation, efficiently downregulating MMP-2 [92,93].

EMT is believed to be the main factor promoting cell migration and invasion in cancer [94] and is also associated with the acquisition of resistance to chemotherapy. TGF-β1 acts as a metastasis inducer, promoting EMT in advanced stages of tumor progression. Jo et al. [48], analyzing human non-small lung cancer cells, demonstrated that KMP strongly inhibited TGF-β1-induced EMT, elevating E-cadherin expression and suppressing the induction of mesenchymal markers. Interestingly, KMP reversed TGF-β1-mediated Snail1 induction. Resistance to chemotherapy is one of the main causes of antineoplastic treatment failure. Combination therapy with sensitizing agents represents a successful approach to suppressing cancer cells and inhibiting the emergence of drug resistance. Within this context, Riahi-Chebbi et al. [95] evaluated the anticancer properties of some polyphenols, particularly KMP, alone or together with 5-FU, in a 5-FU-resistant colorectal cancer cell line (LS174-R). The authors showed that KMP can overturn 5-FU resistance of LS174-R cells through the induction of apoptosis and cell cycle arrest [95]. Furthermore, this agent prevented the production of ROS. Thus, KMP may be a promising chemotherapeutic agent to be used alone or in conjunction with 5-FU to reverse resistance to chemotherapy drugs [95].

Despite the limitations of in vitro models in fully mimicking the native tumor microenvironment and, consequently, in testing effective antineoplastic therapies, significant results have been obtained with the use of KMP in different neoplastic processes. Future studies using 3D in vitro models, which are more representative of tumor complexity, and animal models may strengthen the understanding of the use of KMP in cancer and its different mechanisms of action. In addition, as a dietary flavonoid, KMP has the limitations of rapid metabolization, low solubility, and low bioavailability, which may represent major obstacles to its anticancer effects in vivo. In view of the findings of the in vitro studies, we believe that there is a strong enough justification to explore the potential applications of KMP in in vivo models. Epidemiological studies evaluating the preventive role of KMP in the development of carcinogenesis are also recommended.

Given the in vitro potential of KMP to act deeply at the molecular level, affecting the DNA structure in the inhibition of cancer and the different mechanisms involved in tumorigenesis, further research is needed to establish the exact signaling pathway mediated by tumor-specific KMP. First, it is necessary to study and identify potential biomarkers that can predict the sensitivity of different tumors to KMP. Second, ex vivo experiments are needed to further clarify the antitumor effects of KMP for cancer prevention and treatment, in addition to studies exploring the effects of KMP combined with other drugs. Finally, the preventive role, efficacy, and tolerability of KMP should be evaluated in longitudinal clinical studies to support the clinical application of KMP in the future. Therefore, further research into the development of KMP as a new molecularly targeted agent against cancer is needed.

## 5. Conclusions

Previous reviews have evaluated the effects of KMP on different types of cancer and other pathological processes [96,97]. Nonetheless, this is the first systematic review that evaluates the in vitro effects of KMP on different types of cancer. The results of this systematic review indicate that KMP exerts in vitro inhibitory effects on several carcinogenesis pathways in different solid cancers, particularly the promotion of cell death, inhibition of proliferation, angiogenesis, and EMT, and sensitization of neoplastic cells to chemotherapy with different drugs. Therefore, KMP as an adjuvant drug, combined with other chemotherapeutics, even targeted drugs, may be a more effective approach to improving therapeutic outcomes and the quality of life of patients with malignant neoplasms.

Research into cancer therapies has resulted in the development of natural medications that demonstrate a milder negative impact compared to traditional cancer drugs. The different functions exerted by KMP allow this molecule to act on various targeted proteins to downregulate disease progression. However, their translation from bench to bedside is still an area that needs to be explored. Regarding future directions, investigations should be carried out in animal models into the effects of KMP in neoplastic treatment, evaluating its mechanisms, possible adverse effects, and appropriate dosages for treatment in different types of cancer. As KMP targets multiple pathways, its use in a multi-targeted therapy that provides additive or synergistic chemotherapy effects could be a good option for cancer treatment. When it comes to clinical application, the use of KMP in clinical studies faces several challenges. The bioavailability of KMP, selection of a dosage range for clinical/therapeutic application, possible restrictions on pharmacokinetics, and the development of delivery systems should be the focus of future studies.

## Figures and Tables

**Figure 1 cancers-16-00585-f001:**
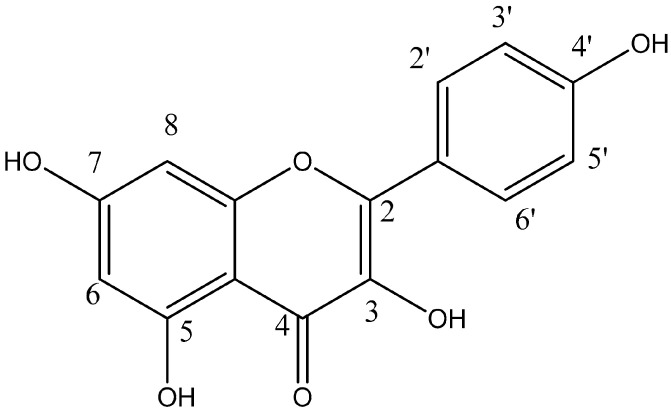
Chemical structure of Kaempferol. Kaempferol is a tetrahydroxyflavone, with the hydroxy groups distributed at positions 3, 5, 7, and 4′.

**Figure 2 cancers-16-00585-f002:**
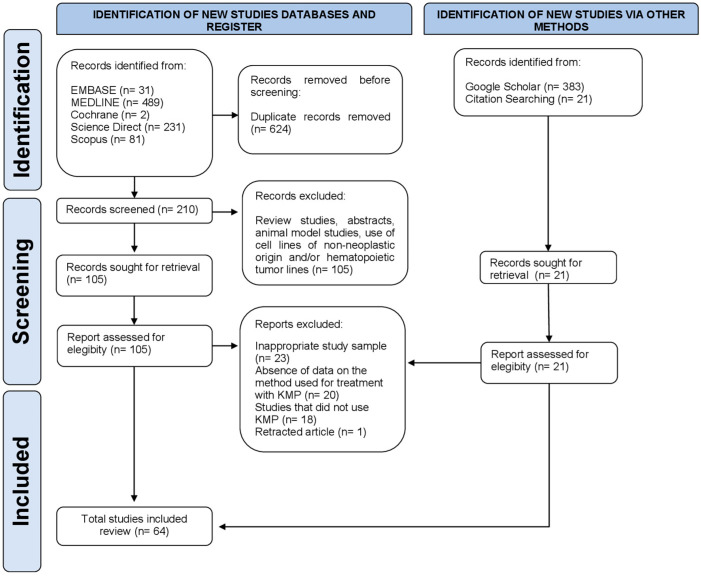
Flowchart of literature search and selection criteria. The study followed the Preferred Reporting Items for Systematic Reviews and Meta-Analyses (PRISMA) guidelines.

**Figure 3 cancers-16-00585-f003:**
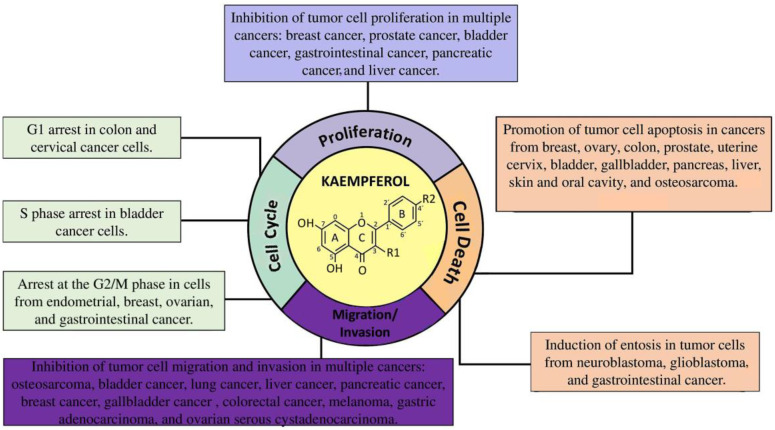
Antineoplastic mechanisms regulated by kaempferol in different types of cancer.

**Figure 4 cancers-16-00585-f004:**
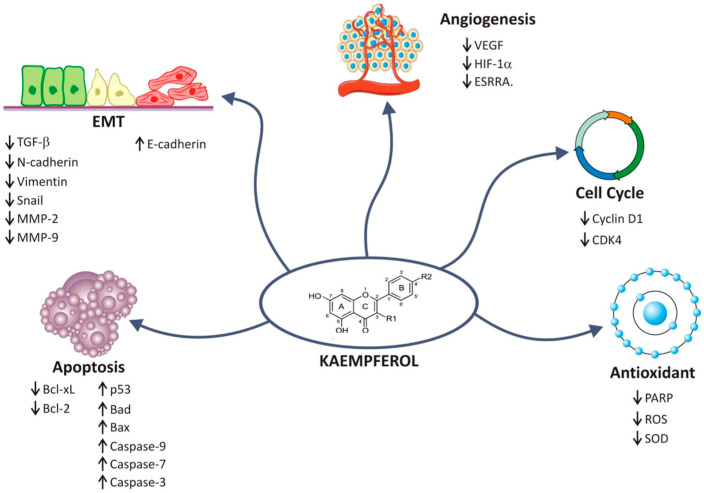
Schematic representation of the pathways and targets affected by kaempferol identified in in vitro cancer studies.

**Table 1 cancers-16-00585-t001:** Main findings, including cell lines, dosage of kaempferol, and time on treatment, distributed by type of tumor.

Cancer Type	Cell Lines	Dosage of KMP	Time on Treatment	References
Bladder cancer	5637, T24, T24L, EJ	10–320 mM	72 h	[46,62]
Bone cancer	U-2, HOB, 143B	25–200 µM	24–48 h	[35,42]
Breast cancer	PMC42, HuTu-80, MDA-MB-453, MCF-7, T47D, MDA-MB-231, HC-11, BT-549, BT474	0.5–200 µM	1–336 h	[23,26,29,30,38,49,50,53,60,65,79]
Cervical cancer	HeLa, SGC-7901	12–100 mM 100–400 μg/mL	24–168 h	[28,44,53,55]
Colon cancer	HT-29, RKO, HCT-116, HCT-8, HcT-116, YB5, KNC, HCT-116, SW480, DLD-1, CT26	0.01–200 µM	24–168 h	[10,12,14,24,32,43,45,59,60,67,76]
Endometrial cancer	HEC-265, HEC-108, HEC-180	36–72 µM	48 h	[56]
Gastrointestinal cancer	AGS, SNU-216, Caco-2, NCI-N87, SNU-638, and MKN-74, MKN-45, MKN28, SGC7901	4–100 µM 100–400 μg/mL	4–168 h	[23,28,33,51,58,79]
Head and neck cancer	SCC-25, SCC-4, SCC-1483), Eca-109, PCI-13, FaDu	20–80 µM	24 h	[36]
Liver cancer	HCC, HepG2, Huh-7, BEL7402 SMMC, Pri-1/-2/-3, SK-Hep-1, Huh7, N1S1, Huh-1, PLC/PRF/5, HLE, HLF, Hep3B	5–200 µM 100–400 μg/mL	24–168 h	[10,22,28,39,44,54,64,68,69,72,77]
Lung cancer	A549, NCIH460, H460, 95-D	5–200 μM 100–400 μg/mL	2–168 h	[22,27,28,44,47,48,53,60,61,70,71,73,80]
Nervous system cancer	IMR32, N2A, U87 MG, U251, U373	5–120 µM	24–96 h	[31,60,64,66]
Ovarian cancer	A2780/CP70, OVCAR-3, A2780/wt, SKOV3IP1	4–160 µM	24–48 h	[34,37,40,41,57,60,79]
Pancreatic cancer	Miapaca-2, Panc-1, and SNU-213, BxPC-3, PANC-1	0–1000 µM	24–336 h	[52,60,75,78]
Prostate cancer	LNCaP, 22Rv1, PC-3, DU145	50 mM	72 h	[25,60,81]
Skin cancer	A375, A431, B16F10	10–400 μg/mL 2–10–80 µM	24–168 h	[10,17,28,60,63]
Others	SGC996, GBC-SD	0–200 μg/mL	24–48 h	[74]

## Data Availability

The data presented in this study are available in this article.

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
