# Peer review of "The Anticancer Potential of Kaempferol: A Systematic Review Based on In Vitro Studies"

_cancers, 2024, doi:10.3390/cancers16030585_

Round 1

Reviewer 1 Report

Comments and Suggestions for Authors

The acronyms MeSH, MAPK, BAD, AKT, STAT, VEGF, TCA cycle, EMT, JNK, EGFR etc. must be explained at the first appearance in the text.

Lines 100-102: please verify the correctness of the information. I did not understand why the in vitro studies were excluded, if the title refers to these studies?

The terms "tissue origin" (line 107) and "effects on expression" (line 109) should be presented more explicitly.

I find the work useful for specialists and interesting from the point of view of the work methodology (the method of selection, analysis and processing of information).

Author Response

We would like to thank you for providing us with timely and valuable comments and suggestions to improve the quality of manuscript. We would also like to thank you for highlighting the quality of our methodological protocol. We have incorporated your suggestions (highlighted in track changes), and below find the point-by-point answers for each of your questions/comments.

The acronyms MeSH, MAPK, BAD, AKT, STAT, VEGF, TCA cycle, EMT, JNK, EGFR etc. must be explained at the first appearance in the text.

We have included a list with the abbreviations and acronyms used in this manuscript, as recommended, but we also spell out the names or short sentences after the first appearance, followed by the abbreviation in parentheses.

Lines 100-102: please verify the correctness of the information. I did not understand why the in vitro studies were excluded, if the title refers to these studies?

We apologize for this mistake; this is clearly a mistake. Instead of in vitro, it should be in vivo. The manuscript has been carefully corrected.

The terms "tissue origin" (line 107) and "effects on expression" (line 109) should be presented more explicitly.

Many thanks for highlighting those important aspects. We have revised this sentence to enhance clarity, and now it reads as: “The authors independently extracted the following data from the included studies using a pre-established form: authors, year of publication, country, tissue origin of cell lines, origin/extraction method of KMP, dosage of KMP, treatment time, effects on biological behavior, effects on expression of biomarkers, effects on cell sensitization to chemotherapy, and main findings of study.”

I find the work useful for specialists and interesting from the point of view of the work methodology (the method of selection, analysis and processing of information).

We really appreciate your encouraging comments about the contribution of our study.

Reviewer 2 Report

Comments and Suggestions for Authors

Review of Coletta and co-workers describes the application of Kaempferol as natural anticancer molecule. The topic of this manuscript is interesting and fits with the target of the Journal. However, before the publication, some issues need to be addressed. 

  • The entire manuscript is divided into sections characteristics of a research article. In general, a review consists of an introduction, sections relative to the literature and conclusions/future perspectives. I strongly suggest to reorganize the entire manuscript, dividing the cited papers by function of the tumor types, as reported in table 1. 

  •  
  • The details of how the literature search has been employed must be removed. 

  •  
  • The authors should discuss if other reviews on the same topic have been published (see for example Neurochem. J. 2023, 17, 247; Critical Rev. Food. Sci. Nutr. 2023, 63, 9580; and many others...), and the relative novelty of this manuscript respect to the previous published 

  •  
  • Conclusion section needs to be improved, including the future perspectives on this field

Author Response

We would like to thank you for the encouraging comments on the achievements and quality of study, and for the suggestions to improve it. We have incorporated your suggestions (highlighted in track changes), and below find the point-by-point answers for each of your questions/comments.

The entire manuscript is divided into sections characteristics of a research article. In general, a review consists of an introduction, sections relative to the literature and conclusions/future perspectives. I strongly suggest to reorganize the entire manuscript, dividing the cited papers by function of the tumor types, as reported in table 1.

Thank you for your careful reading of the manuscript and for those important comments. We initially highlight that the manuscript, as it is a systematic review, follows the structure of the Preferred Reporting Items for Systematic Reviews and Meta-Analyses (PRISMA) guideline, as recommended by the Cancers’ Instructions for Authors.

Regarding the description of the results, this has been extensively discussed by the authors, and your recommendation was considered. The main reason to describe the results by phenotype/biological effect was that KMP effects were repetitively assessed for the same biological processes in the different studies, and the results, in your majority, were quite similar among different tumor types. In this context, the description by tumor type would make the text very repetitive. Moreover, describing by phenotype/biological effect, we show that KMP effects are not tumor-dependent, highlighting its anticancer effects for different types of tumors. With respect, we have chosen to keep the original structure, with description by phenotype/biological effect, but also depicting the main features and findings of the individual studies on Appendix C and the summary of the main findings according to type of tumor on Table 1.

The details of how the literature search has been employed must be removed.

We appreciate and respect your suggestion, but the study followed the PRISMA guideline for systematic reviews and the strategy of literature search is described as recommended in the guideline. With respect, these relevant data were maintained.

The authors should discuss if other reviews on the same topic have been published (see for example Neurochem. J. 2023, 17, 247; Critical Rev. Food. Sci. Nutr. 2023, 63, 9580; and many others...), and the relative novelty of this manuscript respect to the previous published.

This is a very astute comment, and we thank you for giving us a chance to highlight the novelty of this review. Although previous reviews have evaluated the effects of KMP on different types of cancer and other pathological processes, this is the first systematic review focused in vitro effects of KMP on different types of cancer. In the introduction and conclusion, we have reinforced this point.

Conclusion section needs to be improved, including the future perspectives on this field.

In the revised manuscript, we have improved the conclusion by including the future perspectives on application of KMP as an anticancer drug. The following paragraph was incorporated to the revised manuscript: “Research into cancer therapies has been resulting in the development of natural medications that demonstrate a milder negative impact compared to traditional cancer drugs. The different functions exerted by KMP allow this molecule to act on various targeted proteins to downregulate disease progression. However, their translation from bench to bedside is still an area that needs to be explored. Regarding future directions, investigations should be carried out in animal models into the effects of KMP in neoplastic treatment, evaluating its mechanisms, possible adverse effects and appropriate dosages for treatment in different types of cancer. As KMP targets multiple pathways, its use in a multi-targeted therapy that provides additive or synergistic chemotherapy effects could be a good option for cancer treatment. When it comes to clinical application, the use of KMP in clinical studies faces several challenges. The bioavailability of KMP, selection of a dosage range for clinical/therapeutic application, possible restrictions on pharmacokinetics, and the development of delivery systems should be the focus of future studies.”

Reviewer 3 Report

Comments and Suggestions for Authors

This systematic review aims to highlight the potential therapeutic effects of KMP on different types of solid malignant tumors. I think this manuscript does not meet the high criteria of the Cancers journal. They could try another journal. I recommend “reject”.

Here are the points:

·       Authors should have given more details about the references they gave.

·       They exaggerated the searching part for the references for Kaempferol. All researchers do the same procedures for preparing an article. The materials part seems like a lesson for the beginners how to write an article. Briefly, these parts must be removed.

·       There should be more quality figures to express the potential of Kaempferol.

·       The table at the end is also unnecessary.

Comments on the Quality of English Language

There are several grammatical errors.

Author Response

We would like to thank you for your critical evaluation of our study. We are sorry for your recommendation, because the relevance of the manuscript is high and the complexity of review is supported by many studies exploring KMP as a potent anticancer natural agent.

Authors should have given more details about the references they gave.

It is not clear the type of detail about the references that the reviewer is asking for. We have applied a well-defined protocol following the PRISMA 2020 guideline, with clear approaches of search strategy, eligibility criteria, study selection, data extraction and data synthesis, quality assessment, among others. Specifically answering your comment, as recommended by the PRISMA 2020 guideline, information source and search strategy was reported in Materials and Methods, and Appendix A describes the full search strategies for all databases, registers and websites, including any filters and limits used.

They exaggerated the searching part for the references for Kaempferol. All researchers do the same procedures for preparing an article. The materials part seems like a lesson for the beginners how to write an article. Briefly, these parts must be removed.

As recommended by the Cancers’ Instructions for Authors, systematic reviews are strongly recommended to follow the Preferred Reporting Items for Systematic Reviews and Meta-Analyses (PRISMA) 2020 guideline. According to the PRISMA updated guideline, the methods must be detailed and presented as an important section of the article.

There should be more quality figures to express the potential of Kaempferol.

Besides the appendices, we have included Table 1, reporting the study characteristics and main findings, and Figures 1-4, ranging from Chemical structure to pathways and targets affected by Kaempferol identified in in vitro cancer studies. With respect, we have covered the main features explored in this systematic review and we do not feel it is necessary to include more figures in the interest of space and focus.

The table at the end is also unnecessary.

It is not clear if the criticism is about the Appendices or is specifically about the Appendix D (last Table), which depicts the assessment of risk of bias, a central component of systematic reviews. All Appendices are required by the PRISMA 2020 guideline.

Reviewer 4 Report

Comments and Suggestions for Authors

my comments are in the pdf attached

Comments on the Quality of English Language

major revision

Author Response

We thank the reviewer for his thoughtful comments and suggestions. We have revised our manuscript to address those comments, most of which have been incorporated into the revision (alterations are highlighted in the text).

Commented [IB1]: Provide list of abbreviations after this.

We have included the abbreviation list as requested.

Commented [IB2]: So add the research gap and why you focused on only Kaempferol? That is what the readers need to know?

Thank you for your relevant question. As we have mentioned in both introduction and discussion, emerging data highlight the biological and therapeutic properties of KMP, a major active tetrahydroxyflavonoid found in several medical plants. Although there are studies, including few reviews, exploring KMP potential in cancer treatment, no systematic reviews have focused on summarizing the in vitro molecular mechanisms and pathways targeted by KMP in tumor cells. Our systematic review indicates that KMP promotes the death of neoplastic cells, and inhibits proliferation, migration, invasion, angiogenesis, and EMT in different tumors. Moreover, KMP sensitizes tumor cells to chemotherapy with different drugs. Those aspects underscore the relevance of this systematic review, which highlights the potential and critically analyzes the gaps in knowledge that limit the use of KMP in cancer treatment. We combined the answers to your comments IB2 and IB3, and incorporated them into the revised manuscript.

Commented [IB3]: Talk about the novelty of this review before add this statement.

Thank you so much for this very relevant suggestion. We have incorporated the following sentence into the revised manuscript: “There is an extensive literature exploring KMP potential in cancer treatment, but no systematic synthesis of in vitro studies elucidating the molecular mechanisms and pathways targeted by KMP is available. In this systematic review, we sought to summarize the available literature on the in vitro anticancer effects of KMP, discussing the pathways and molecular mechanisms regulated by KMP and highlighting its potential as a natural anticancer drug. We also critically analyzed the gaps in knowledge that limit the use of KMP in cancer treatment.”

Commented [IB4]: A bibliometric analysis of the growth of importance of this topic over the years (past 5 or 10 years) must be done (using Web of Science), or lens.org, and a VOS viewer and a plot with an associated explanation must be added.

This is a very astute and relevant comment. However, our systematic review focuses on qualitative synthesis and critical analysis of existing literature rather than quantifying scientific output. With due respect, we believe that a bibliographic analysis is out of scope and would add limited value to the present study.

Commented [IB5]: Provide insights on future studies that needs to be done.

As recommended, in the revised manuscript, we have improved the conclusion by including the requirements before application of KMP as an anticancer drug. The following paragraph was incorporated to the revised manuscript: “Research into cancer therapies has been resulting in the development of natural medications that demonstrate a milder negative impact compared to traditional cancer drugs. The different functions exerted by KMP allow this molecule to act on various targeted proteins to downregulate disease progression. However, their translation from bench to bedside is still an area that needs to be explored. Regarding future directions, investigations should be carried out in animal models into the effects of KMP in neoplastic treatment, evaluating its mechanisms, possible adverse effects and appropriate dosages for treatment in different types of cancer. As KMP targets multiple pathways, its use in a multi-targeted therapy that provides additive or synergistic chemotherapy effects could be a good option for cancer treatment. When it comes to clinical application, the use of KMP in clinical studies faces several challenges. The bioavailability of KMP, selection of a dosage range for clinical/therapeutic application, possible restrictions on pharmacokinetics, and the development of delivery systems should be the focus of future studies.”

Round 2

Reviewer 2 Report

Comments and Suggestions for Authors

manuscript can be published

Author Response

The authors would like to thank the reviewer for all careful, constructive and insightful comments in relation to this work.

Reviewer 3 Report

Comments and Suggestions for Authors

Authors indicated that Kaempferol exerted in vitro inhibitory effects on several carcinogenesis pathways in different solid cancers, particularly the promotion of cell death, inhibition of proliferation, angiogenesis and EMT, and sensitization of neoplastic cells to chemotherapy with different drugs. The revised form is better, but there is still a missing point, which is not answered well by authores:

In this comment :“Authors should have given more details about the references they gave.”: It is obviously requested from authors to give more specific data about studies they mentioned. For example: “Zhang et al. [81], analyzing three different prostate cancer cell lines (22Rv1, PC-3, and DU145), showed that KMP exerts similar effects in terms of reducing cell proliferation but at different points depending on the cell line, arresting the cell cycle at G1 in 22Rv1 and at S/G2 in PC-3.” It is not clear how KMP caused these effects and with what value. There is only simple result explanation. This is a review study it must be more detailed. In whole manuscript, the references were mentioned like that, which is not detailed for a review study.

Author Response

Thank you for this important comment. As suggested, we have detailed the results of some of the studies bringing key effects of kaempferol.

Reviewer 4 Report

Comments and Suggestions for Authors

Accept 

Comments on the Quality of English Language

Minor revision 

Author Response

The authors would like to thank the reviewer for his comments that certainly improved the manuscript. Regarding the quality of text, a native English-speaker, with science-writing expertise, has again gone over the manuscript and few mistakes were corrected.

Round 3

Reviewer 3 Report

Comments and Suggestions for Authors

The authors have satisfactorily addressed my concerns and it can be accepted.